# Community emergency management policy in China using a policy text tool

**Hehua Du**[1]*, **Xingyue Wang**[2]

1 Department of Emergency Management, Xihua University, Chengdu, China, 2 Department of Law and Sociology, Xihua University, Chengdu, China

* duhehuagl@163.com

## Abstract

Systematically analyzing the current status and problems of the Community Emergency Management Policy (CEMP) system in China and proposing practical suggestions that are conducive to subsequent policy formulation and improvement are important for improving community emergency management capabilities and levels and promoting sustainable community development. Based on the two-dimensional analysis framework of "policy tools–policy objectives", this paper draws from content analysis and social network analysis, quantitatively analyzing the CEMP texts at the national level from 2004 to 2024. The results show that the CEMP system in China has essentially taken shape, but there are still some problems, such as the uneven use of policy tools, the unbalanced internal structure of policy tools, the large distribution gap of policy objective elements, and the adaptability between policy tools and policy objectives, which needs to be improved. Given these results, we suggest 1) Appropriately increasing the frequency of using demand-type policy tools and effectively optimizing the internal structure of the three main types of policy tools; 2) Comprehensively deepening the reform of CEMP objectives; and 3) Reasonably improving the adaptability between policy tools and objectives.

## 1. Introduction

With the economic and social transformation of China, its urbanization is accelerating, which has led to the increasing demand of residents for community public safety. In 2020, the COVID-19 pandemic swept the world, causing great challenges to the community emergency management in China. During the emergency management of the COVID-19 pandemic, a series of problems were identified, including weak emergency management ability and imperfect systems and mechanisms of grassroots communities in China [1].

Emergency management is an important means for the government to control, mitigate and eliminate serious social hazards caused by emergencies and to maintain national security, public safety, environmental security and social order. As the basic unit of governmental public governance, the community is at the forefront of emergency prevention and management and plays a safeguarding role in emergency management and social governance. Therefore, the Chinese government has attached great importance to building and improving community emergency management capacity and has issued several strategic plans and policy documents to provide procedural and institutional safeguards for

**Funding:** This research is supported by the project of "Wisdom Emergency Management Key Laboratory" (No. 2023ZHYJGL-4) of the Sichuan Provincial Key Laboratory of Philosophy and Social Sciences and the Youth Fund for Humanities and Social Sciences Research of the Ministry of Education of the People's Republic of China (No. 23YJCZH051). The author is the person in charge of the first project and a participant in the second project. Note: The funders had no role in study design, data collection and analysis, decision to publish, or preparation of the manuscript.

**Competing interests:** The authors have declared that no competing interests exist.

community emergency management. At the same time, Jinping Xi, the general secretary of the Communist Party of China, in his report to the 19th National Congress of the Communist Party of China (CPC), stated that it was necessary to "improve the national security system and strengthen national security capacity building". National policy can provide important support for the community to manage emergencies [2] and proactively guide the public and enterprises to participate in orderly emergency relief and postdisaster recovery and reconstruction and enhance social cohesion. However, when dealing with emergencies, CEMPs in China have significant problems, such as a lack of functions and inefficient implementation [3]. Hence, we attempt to systematically analyze the overall characteristics of China's CEMP system, summarize the various types of policy tools and their drawbacks, and propose practical improvements for subsequent policy formulation and improvement. This will help improve the community's emergency management capacity at the policy level, enhance the existing policy system, promote efficient policy outputs, and energize the community's sustainable development.

## 2. Literature review

With the frequent occurrence of natural disasters, accidents, catastrophes and public health incidents, the community, as the "first responder" to prevent and control disaster events [4], has received extensive attention in research on community emergency management. According to existing research, most studies focus on three topics: community resilience, ways to improve community emergency response capability and evaluations of community emergency response capability.

First, several scholars have investigated the link between community resilience and emergency response. Williams et al. explored the dynamic relationship between emergency management and resilience on the basis of a literature review [5]. Lam et al. argued that effective emergency management can promote community resilience, which should focus on the whole process of managing emergencies [6]. Moreno et al. suggested that communities are not only disaster victims but also active actors and that resilience practices, such as community awareness, organization, cooperation, and social capital, contribute to the recovery of the entire community. Resilience reduces the vulnerability of residents to disasters by activating local resources [7], with organizations the most important resilient force [8]. Latvakoski et al. argued that in the case of disaster damage to cyberinfrastructure, applying innovative technologies can increase residents' disaster risk awareness and community resilience [9].

Second, research on the path of community emergency response capacity enhancement, organizing emergency response drills [10], strengthening the training of community emergency response teams [4], promoting cooperation among community residents [11], and shaping community leaders [12] can enhance community emergency response capacity. Moreover, improving community participation mechanisms is important for communities to address disasters, and the government should establish and fund community emergency response teams to help the government engage in dialogue with citizens and promote collective coping strategies to improve the overall emergency response capacity of the community [13]. Hu et al. argued that community response to public health emergencies can be improved by coordinating the relationship between the government and various organizational stakeholders, emphasizing community participation in prevention and control, strengthening public opinion guidance and public communication, and promoting international cooperation [14]. Wu confirmed that emergency response education, social participation, expert panels, and community relations are important factors influencing the sustainable development of disaster-resistant communities [15].

Third, to evaluate community emergency management capacity, Wang et al. established a model based on hierarchical analysis and the superior-differential distance method, evaluated community emergency management capacity, and proposed suggestions for improvement, such as increasing material reserves, strengthening community emergency drills, and conducting censuses of emergency shelters [16]. Zhang et al. noted that a city's emergency response capacity can help the government better cope with sudden-onset disasters and applied evidence theory to evaluate urban emergency response capacity [17]. Community fire emergency response capability is an important representation of the government's emergency response capability. Wu et al. constructed an evaluation system of urban community fire emergency response capability on the basis of fuzzy hierarchical analysis and demonstrated the practicability and efficiency of the system through examples [18]. Chen et al. used the above research method to evaluate the level of emergency response capability of China's fire stations [19]. Ju et al. proposed a hybrid fuzzy method composed of a fuzzy analytic hierarchy process and binary fuzzy language method to evaluate emergency response ability. They verified the efficiency and feasibility of this method [20].

Research on the resilience, capacity improvement path and capacity evaluation of community emergency management has led to many useful conclusions; however, there is still room for further research. First, prior studies have focused mainly on enhancing the capacity of community emergency management and its evaluation; however, few studies have carried out in-depth analysis of CEMPs. Second, there is a lack of research evaluating CEMPs from the viewpoint of policy tools, which, as a governmental governance strategy, directly affects policy implementation[21]. Quantitative analysis of policy texts using policy tools has recently become the mainstream method of policy analysis, and scholars from different research fields have conducted in-depth studies from the perspective of policy texts on policy topics such as environmental governance [22,23], high-speed rail [24], energy [25], and medical policy [26]. However, research has not yet focused specifically on CEMP. Therefore, to fill the research gap from the policy tool perspective, this paper utilizes the content analysis method, which is based on the two-dimensional analysis framework of "policy tools–policy objectives". Systematic research on policy texts from the dimensions of policy tools and policy objectives is conducted, revealing the urgent, in-depth problems in current CEMPs. On this basis, this study proposes adjustments to the current CEMPs, which have theoretical and practical value for future improvement of emergency management systems in China.

## 3. Research framework and policy text selection

### 3.1 Research framework

Drawing on the two-dimensional analysis framework of "policy tools–policy objectives", this paper uses content analysis and social network analysis to quantitatively study CEMP texts in China. It expands the analysis framework, method category and policy suggestions of CEMP evaluation in China and is innovative. First, based on the official websites of the central government and functional departments of China and the Peking University magic weapon database, 87 policies related to community emergency management were collected as the research objects. Second, the release time of CEMPs in China is analyzed, and the collinear network of CEMP text keywords is outlined using the social network analysis method to reveal the central keywords. Third, within the two-dimensional analysis framework of "policy tools–policy objectives", the content analysis method is used to quantitatively analyze the policy texts. Finally, according to the current situation of using policy tools and target planning of community emergency management in China, suggestions for policy optimization are proposed. The research process is shown in Fig 1.

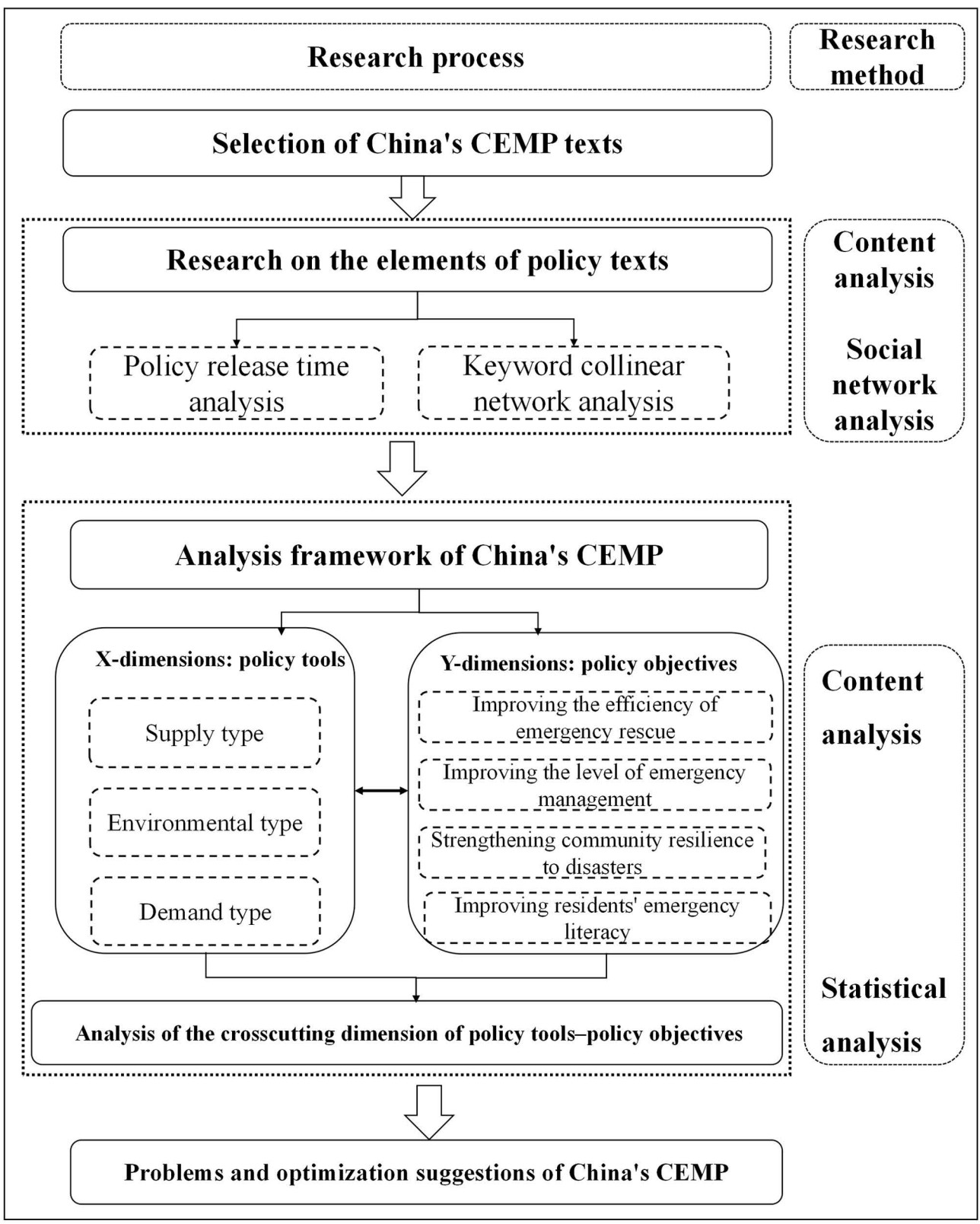

**Fig 1. Analysis flow chart.**

## 3.2 Policy text selection

In searching for and selecting China's CEMPs, we adopt the following procedure. First, because local policies are generally a refinement or continuation of national policies, we focus on nationally applicable CEMPs issued by the central government and functional departments. Second, we search for officially published documents, such as laws and regulations, strategies, plans, rules, methods, notices, and opinions related to community emergency management using the official websites of the central and functional departments and the magic weapon legal database of Peking University as text data sources. We take "community emergency management", "grass-roots emergency", "community emergency", "community emergency drills" and "community emergency rescue" as keywords. If the policy has been revised, the latest revision will be adopted, and the invalid policy text will be deleted. Finally, after the 2003 SARS outbreak in China, emergency management was very important to the government and the public, and the comprehensive strengthening of emergency management started. Therefore, this paper selected policy texts from the 2004–2024 period. Through systematic retrieval and selection, 87 policies highly related to community emergency management were finally collected, as shown in Table 1.

## 3.3 Research methodology

This paper uses the content analysis and social network analysis methods to quantitatively study CEMP texts.

Content analysis combines quantitative and qualitative content analysis and is widely used in various types of policy text research [27,28]. This method can be used to construct unstructured content, divide text content into specific categories, and calculate the frequency of content elements in categories to describe content characteristics and quantitatively analyze qualitative issues. That is, "quantitative" analysis is used to discover the "qualitative" characteristics and generate objective, systematic, quantitative characteristics [29]. In this paper, content analysis is used to encode national CEMP texts from 2004 to 2024. CEMPs in China are quantitatively analyzed using the two-dimensional analysis framework of "policy tools–policy objectives".

Drawing from graph theory, social network analysis uses quantitative tools to visualize the interactions between individuals [27]. It is used mainly to analyze the relationship structure and attributes of social networks. Social network analysis can be applied in many fields. The collinear keyword network clearly shows the relationships between feature members and

**Table 1. Texts of CEMP at the national level (partial).**

| NO. | Policy Name | Department | Release Time |
|---|---|---|---|
| 1 | <Opinions on the implementation of the outline for comprehensively promoting administration according to law> | the State Council | 2004 |
| 2 | <Notice on printing and distributing the overall implementation plan of emergency management science popularization and education work> | the State Council | 2005 |
| ...... | ...... | ...... | ...... |
| 56 | <Opinions on Further Playing the Role of Emergency Broadcasting in Emergency Management> | Emergency management department, etc. | 2020 |
| 57 | <Opinions on Improving and Perfecting the Comprehensive Service Functions at the Village Level> | Emergency management department, etc. | 2020 |
| ...... | ...... | ...... | ...... |
| 86 | <Opinions on Further Improving the Emergency Management Ability at the Grass-roots Level> | the General Office of the Central Committee of the CPC | 2024 |
| 87 | <People's Republic of China (PRC) Emergency Response Law> (revised in 2024) | the National People's Congress Standing Committee (NPCSC) | 2024 |

the keyword position structure [30], which helps combine the "micro" relationship network between individuals with the "macro" structure of a large-scale social system. Using ROS-TCM6.0, this paper draws the keyword colinear network of China's CEMP texts, reveals their key points through keyword combination, and explores keyword key points and correlations on the basis of the network node position and connecting line.

This paper also uses mathematical statistics and, with the statistical analysis function of data analysis software, such as Excel, completes the statistical work of the number of texts, the distribution of policy tools and the target characteristics of CEMPs. It intuitively reflects the basic characteristics of CEMPs through data and proportional relationship diagrams.

## 4. Construction of the policy analysis framework

In addition to the rationality element that needs to be considered in selecting policy tools, the importance of the value goal element should not be overlooked [31]. A reasonable policy structure should reflect the organic combination of policy tools and value objectives. To clarify the aspects in which the CEMP has improved emergency management and standardized emergency response activities, as well as those areas in which it does not provide effective policy support, it is necessary to consider the policy objectives of community emergency management in depth to reasonably analyze policy too. Therefore, this paper comprehensively explores the dual perspectives of policy tools and objectives utilizing CEMP tools in China.

### 4.1 X-dimensions: policy tools

Policy tools are effective policy interventions that decision-makers and practitioners use to solve policy problems and achieve policy goals in a given policy environment [32]. As the basic unit of public policy analysis, studies divide policy tools into various types according to different classification standards. For example, Howlett et al. focused on the strength of state intervention for dividing policy tools into three types: voluntary, hybrid and mandatory [33]. Rothwell et al. classified the policy tools into demand, supply, and environment types [34], each of which can be subdivided into specific policy tools. Lorraine et al. established a five-classification framework to categorize policy tools into command, incentive, capacity building, systemic change, and symbolism and exhortation [35]. Rothwell et al.'s criteria for classifying policy tools not only focus on the degree of government intervention but also comprehensively consider the means of intervention, such as the factors acting on policy supply, the influence exerted by the environment, and the demand factors acting on the market and society. Their further classification of subpolicy tools is also more in line with the dimensions of real policy formulation; thus, it is the most frequent dimension of policy tool analysis adopted by scholars.

Combined with the characteristics of community emergency management in China, this paper adopts Rothwell's criteria for categorizing policy tools, dividing community emergency management policy tools into three dimensions: demand, supply and environment. Among them, supply-type policy tools refer to the government's "top-down" modernization of the community emergency management system through financial support, talent training, technical support, etc., emphasizing the "driving role". Environmental policy tools are mainly used to support community emergency management activities to indirectly realize policy objectives, namely, target planning, strategic measures and regulatory control. Demand-type policy tools are concerned with promoting the community emergency management capacity and efficiency of the market and social demand, mainly through government purchasing, international exchanges and other ways to unite social capital to jointly promote community emergency management, emphasizing the "pull" role. The specific contents are shown in Table 2.

**Table 2. Division of CEMP tools.**

| Tool type | Name | Meaning |
|---|---|---|
| Supply-type | Capital investment | Government financial inputs, financial subsidies around community emergency management. |
| | Talent cultivation | Government develops professional emergency response personnel and teams for the community. |
| | Material support | Material security system constructed by the government for community emergency management. |
| | Perfect facilities | Government provides infrastructure and shelters for the development of community emergency management. |
| | Information support | Government support through scientific and technological means such as big data and information platform construction |
| | Public services | Basic security services, counseling services, resident care services, etc. provided by the government. |
| Environment-type | Target planning | Master plans and macro-objectives proposed by the government to regulate the development of community emergency management. |
| | Strategic measures | Government achieves its goals through measures such as strengthening management, evaluation and monitoring, staff incentives and accountability. |
| | Regulatory control | Government needs to develop a mandatory system of laws, regulations and sectoral rules related to community emergency response. |
| | Science and education promotion | Government's emergency science education and publicity and guidance activities for community residents. |
| | Plans and drills | Government plans and programs, emergency drills for community emergencies. |
| | Public opinion guidance | Government creates a favorable public opinion environment for the positive and prudent handling of public emergencies |
| | Financial taxation | Government incentives for community emergency-related stakeholders through tax and financial instruments. |
| Demand-type | International exchange | Government learns from the advanced experience of foreign communities in emergency response by strengthening external cooperation and exchanges. |
| | Sectoral collaboration | Government departments promote community emergency management through mutual collaboration. |
| | Social participation | Government encourages community residents, enterprises, social organizations and others to participate in community emergency management. |
| | Government purchases | Government departments prioritize the procurement of products and services needed for community. |
| | Leading demonstrations | The government promotes the implementation, demonstration and promotion of emergency demonstration areas and industries. |

## 4.2  Y-dimensions: policy objectives

Policy objectives are the directions and results that a CEMP is expected to achieve, which can be regarded as the basic premise of policy formulation and implementation [36]. Policy objectives not only provide directional guidance for allocating policy tools but also provide the core evaluation criteria for policy implementation. According to the content of policy texts and the characteristics of community emergency management, this paper identifies four policy objectives: improving the efficiency of emergency rescue, improving the level of emergency management, strengthening community resilience to disasters, and improving residents' emergency literacy. Table 3 presents the details.

## 4.3  X-Y two-dimensional analysis framework construction

By combining the policy tools dimension with the community emergency management objectives, the current study aims to show the Chinese government's interventions in community emergency management from an instrumentalized perspective. Moreover, we explore the objectives achieved by the CEMPs from an outcome perspective to effectively articulate the policy tools and objectives and to more comprehensively analyze the CEMPs. Therefore, this paper adopts a two-dimensional analytical framework that combines policy tools and objectives, as shown in Fig 2.

**Table 3. Distribution of CEMP objectives.**

| Policy objectives | Sub-category | Meaning |
|---|---|---|
| Improving the efficiency of emergency rescue | Upgrading emergency response techniques | Enhance the community's technical capacity for emergency response by promoting the construction of emergency information platforms and innovation in emergency response technology. |
| | Improving monitoring and early warning | Promote community risk monitoring and early warning through the establishment of monitoring networks and the development of monitoring equipments. |
| | Building emergency response forces | Build a network of community emergency response forces by strengthening talent and team building and mobilizing participation. |
| | Improving the safeguard system | Improvement of the community emergency protection system through increased funding, public services and financial means. |
| Improving the level of emergency management | Improving institutional mechanisms | Improvement of the community emergency response system mechanism by strengthening institutional mechanisms and implementing measures. |
| | Strengthening supervision and management | Improvement of community emergency management and supervision by strengthening organizational guidance and management and implementing strategic measures. |
| | Promoting integrated planning | Promote integrated community emergency planning through the development of various long- or short-term goals. |
| | Sounding regulations and preplanning | Improvement of various legal systems for emergency response and enhancement of the government's capacity to deal with emergencies. |
| | Implementing treatment rewards | Implementing personnel treatment and incentives through recognition and awards, and placement preferences. |
| Strengthening community resilience to disasters | Strengthening synergies | Enhance disaster response capacity through various types of cooperation and exchanges. |
| | Improving resource reserves | Improve disaster resilience by promoting community stockpiles of materials and resources. |
| | Improving facilities and equipments | Improve community resilience by establishing and improving various types of community emergency infrastructure equipment and shelters. |
| | Promoting demonstration exercises | Improve community resilience through the establishment of model communities, model emergency industries, and various exercise programs. |
| Improving residents' emergency literacy | Mastering emergency knowledge | Promote the acquisition of emergency-related knowledge by residents through science education and publicity and guidance. |
| | Building public awareness | Build public awareness among residents through relevant activities and policy interpretation. |

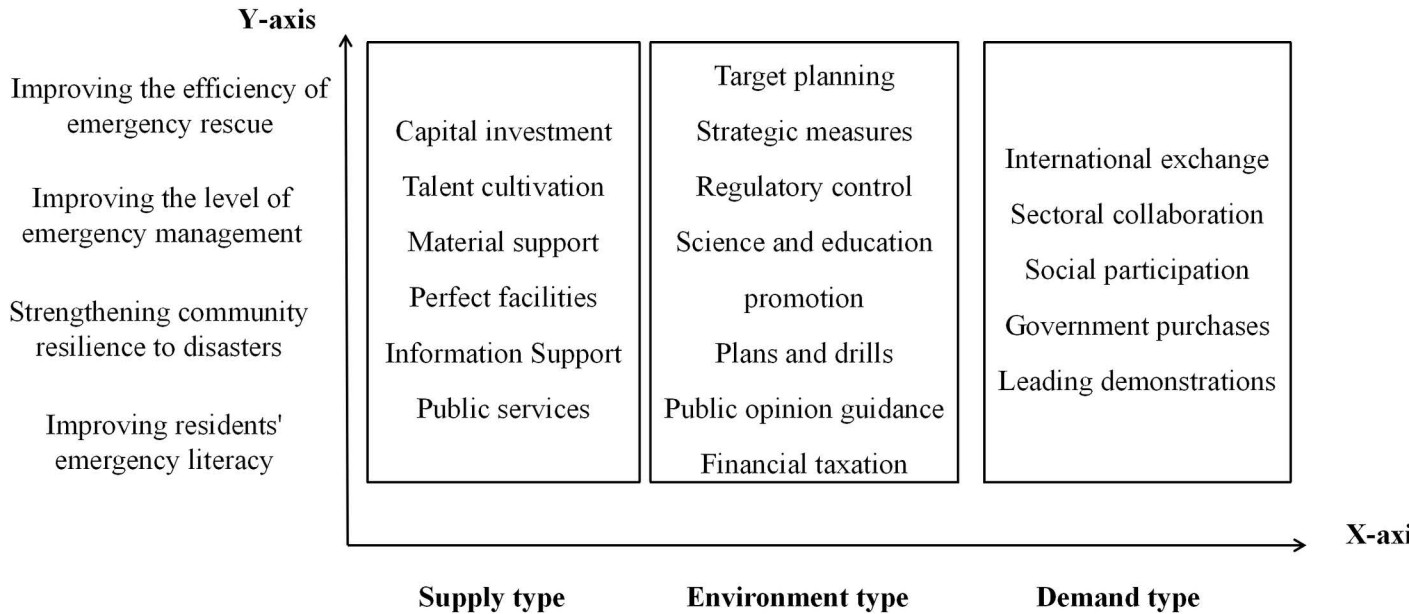

**Fig 2. Framework diagram for two-dimensional analysis of CEMPs.**

Drawing on this framework, we import the 87 policy texts into Nvivo12 qualitative analysis software. The contents related to "community emergency response" are searched and coded according to the rule of "policy number–chapter number–analysis unit number–content sequence number". For example, 1-1-1-1-1 indicates the specific content of the first article of the first content analysis unit of the first chapter of the first policy text. The coding process followed the principle of nondisaggregation; if the specific content embodied multiple policy tools, the coding was repeated to ensure the accuracy and scientific validity of the research findings. Finally, 777 codes were obtained and categorized into the corresponding dimensions of policy tools and policy objectives, as shown in Table 4.

## 5. Quantitative analysis of CEMP texts

### 5.1 Policy release time analysis

From 2004 to 2024, the number of CEMPs introduced in China increased, as shown in Fig 3. China's CEMPs present a staged change divided into three stages based on the time nodes of the change in the number of policies:

In the beginning of the exploration stage (2004–2007), since SARS in 2003, the Chinese government and public began to realize the importance of emergency management; thereafter, the emergency management system developed toward institutionalization, scientification and standardization, with "one case, three systems" at the core. In this stage, the number of policy texts increased, peaking in 2007.

In the development and enhancement phase (2008–2017), various serious natural disasters, such as the Wenchuan earthquake and the southern freezing disaster, posed considerable challenges to community emergency management practices. In 2012, the 18th National People's Congress opened a new period of construction of an emergency management system that was strategically guided by overall national security, resulting in the steady development of CEMP.

**Table 4. Example of text coding for CEMPs.**

| NO. | Name | Content | Encoding | Policy tool | Policy objective |
|---|---|---|---|---|---|
| 1 | <Opinions on the implementation of the outline for comprehensively promoting administration according to law> | Establish and improve all kinds of early warning and emergency laws... properly handle all kinds of emergencies | 1-2-3-1 | Regulatory control | Sounding regulations and preplanning |
| | | Strengthen the training and drills of emergency plans... | 2-2-1-2 | Plans and drills | Sounding regulations and preplanning |
| ... | ... | ... | ... | ... | ... |
| 23 | <Notice on recommending the fourth batch of national comprehensive disaster reduction demonstration community candidate units> | Promote the community to carry out activities such as general survey of hidden dangers, comprehensive drills on disaster prevention and reduction, knowledge and skills training on disaster prevention and reduction, volunteer service on disaster reduction, and publicity on disaster prevention and reduction. | 23-1-2 | Science and education promotion | Mastering emergency knowledge |
| | | Mobilize every family and member of the community to pay attention to all kinds of disaster risks around them, enhance their awareness and skills in preventing and responding to disaster risks, and actively participate in disaster prevention and mitigation and emergency management. | 23-1-3 | Social participation | Building public awareness |
| ... | ... | ... | ... | ... | ... |
| 87 | <People's Republic of China (PRC) Emergency Response Law> (revised in 2024) | Township people's governments, neighborhood offices and conditional residents' committees and villagers' committees can establish grassroots emergency rescue teams to carry out emergency rescue in a timely and nearby manner. | 87-3-1 | Talent cultivation | Building emergency response forces |
| | | The state supports urban and rural community organizations to improve the emergency work mechanism... | 87-5-1 | Strategic measures | Improving institutional mechanisms |

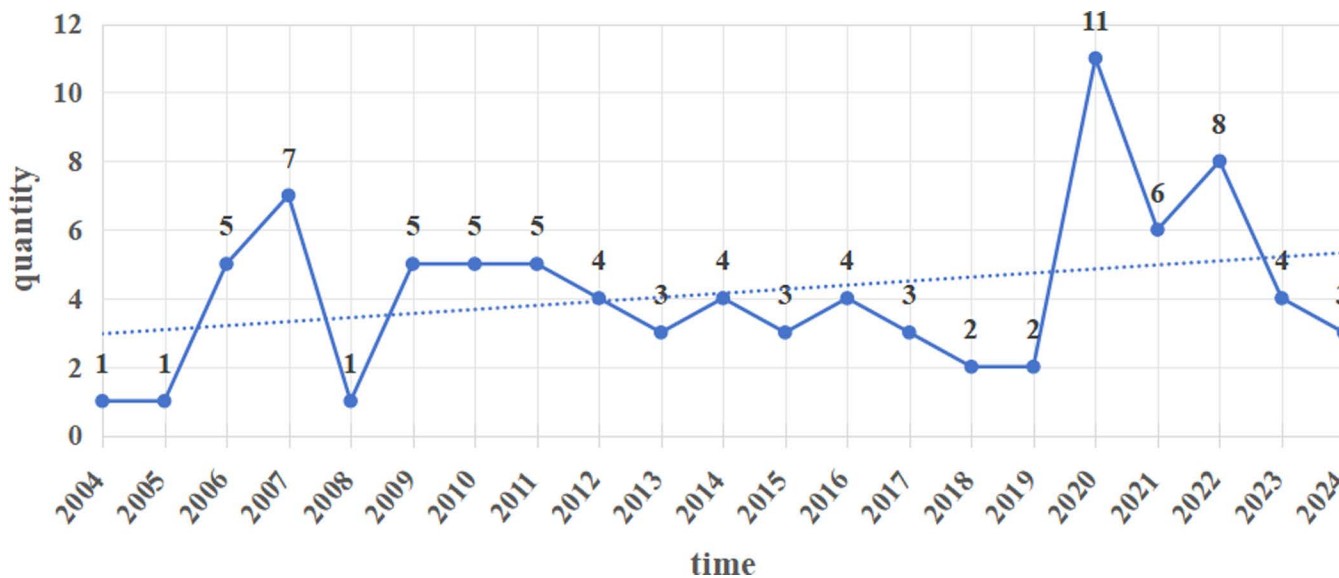

**Fig 3. Text publishing timeline.**

In the improvement and deepening phase (2018–present), since the 19th National Congress in 2017 proposed promoting a downward shift in the center of gravity of social governance and accompanied by the official establishment of China's Ministry of Emergency Management (MEM) in 2018, China's emergency management system has undergone holistic and systematic change, and the CEMPs have entered a period of deepening. Moreover, with the outbreak of the COVID-19 epidemic in 2020, grassroots communities have been at the forefront of preventing and managing disaster events. The Chinese government has successively issued several strategic plans and policy documents to standardize and incentivize community emergency management practices, increasing the number of policies to the highest level.

### 5.2 Keyword colinear network analysis

The extraction and frequency statistics of keywords, the core content of CEMP texts, are helpful for mining the key content information of policy texts. The colinear characteristics of keywords further reflect the relationships among policy texts. ROSTCM6.0 is used to extract the top high-frequency words of CEMPs, construct a keyword co-occurrence matrix, and finally form a colinear network of CEMP keywords, as shown in Fig 4.

### 5.3 Dimensional analysis of policy tools

On the basis of the three types of policy tools, the distribution of policy tools in China's CEMP was obtained on the basis of the content code categorization in Table 2 (see Fig 5). China has comprehensively utilized three types of policy tools—supply, environment and demand types—to support community emergency management. However, there are still some differences in the use of these policy tools: among them, the environmental policy tools are most widely adopted, accounting for 49.29%; the supply type policy tools have the second highest utilization rate, accounting for 33.98%; and the demand type policy tools have the lowest utilization rate, accounting for only 16.73%. The Chinese government prefers policy tools in community emergency management, forming a "supply-type + environment-type" policy tool distribution pattern, with a serious shortage of demand-type

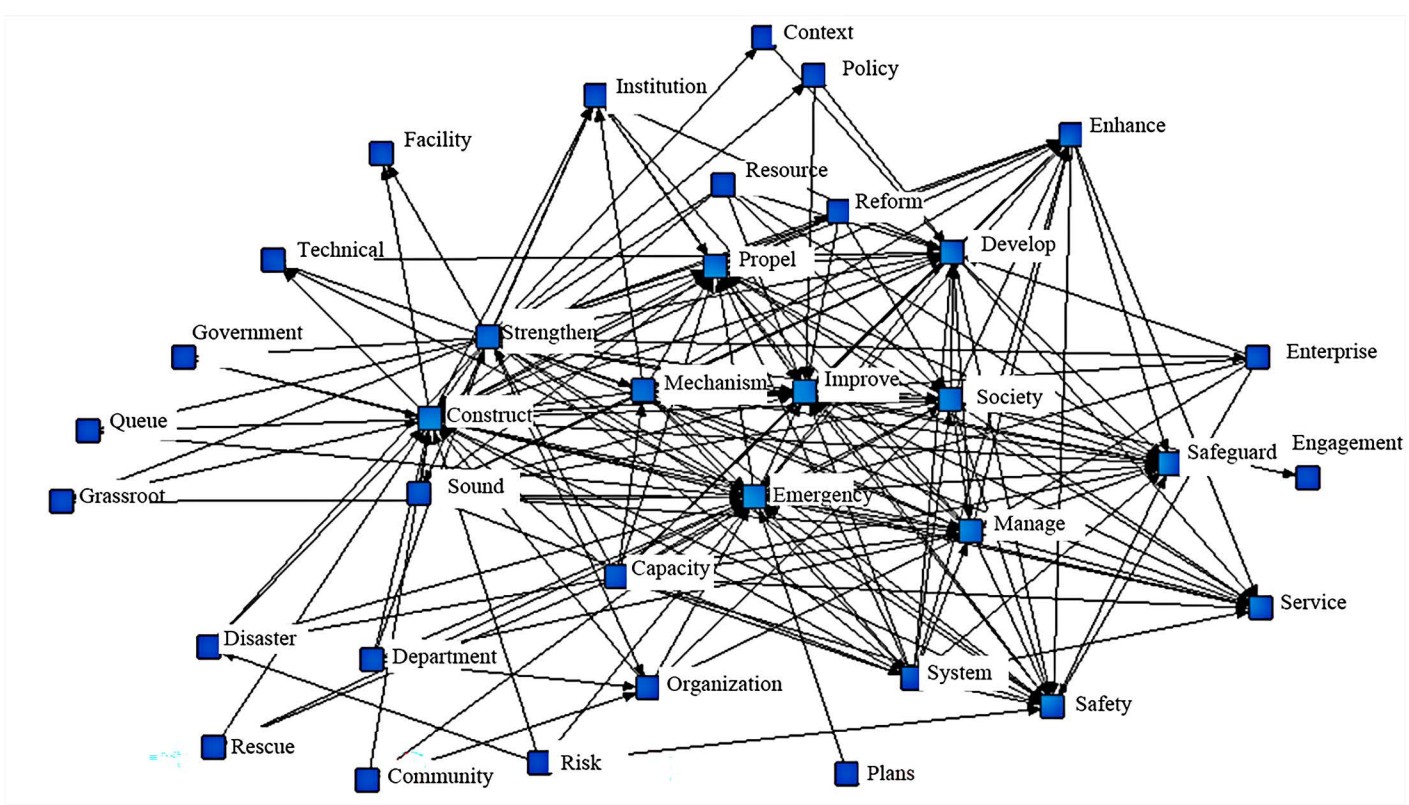

**Fig 4. Colinear network of keywords for CEMPs.** The node position of the keyword colinear network represents its centrality, and the thickness of the connecting lines reflects the relevance of keywords. In Fig 4, the keywords, namely, emergency response, management, capacity, mechanism, construction, development, and society, are focused at the center of the network, presenting an evident correlation with other keywords. The above keywords play important roles in the community emergency management network. As far as the collinear network diagram of CEMP keywords is concerned, China's CEMPs constitute a complex network system that includes an emergency mechanism, emergency capacity, emergency management, an emergency system and other elements, and the interaction of different keywords promotes the development of CEMPs.

policy tools. Most tend to use direct measures to promote community emergency management by providing required human resources, infrastructure, information, institutional support and other resources. Therefore, policy-makers should increase the frequency of the use of demand-type policy tools. First, the government should further delegate power to grassroots organizations, improve the autonomy of emergency management in grassroots communities, establish a mechanism for selecting community emergency management to undertake government projects, provide sufficient and stable material guarantees, actively allocate market resources effectively, and encourage communities to manage resources more effectively [37]. Second, policy-makers should strengthen the construction of an international exchange system of community emergency management [14], actively learn from foreign advanced experience and integrate it with local emergency management methods. Third, policy-makers should improve the cooperation mechanism of community emergency management departments and promote the coordinated governance of the government, enterprises and communities [38].

Among the supply-type policy tools, the internal distribution is relatively balanced, indicating that the Chinese government uses various supply-type policy tools comprehensively in response to public emergencies in the community. First, "talent cultivation" has been used most frequently, accounting for 8.62%, indicating that cultivating professionals in community

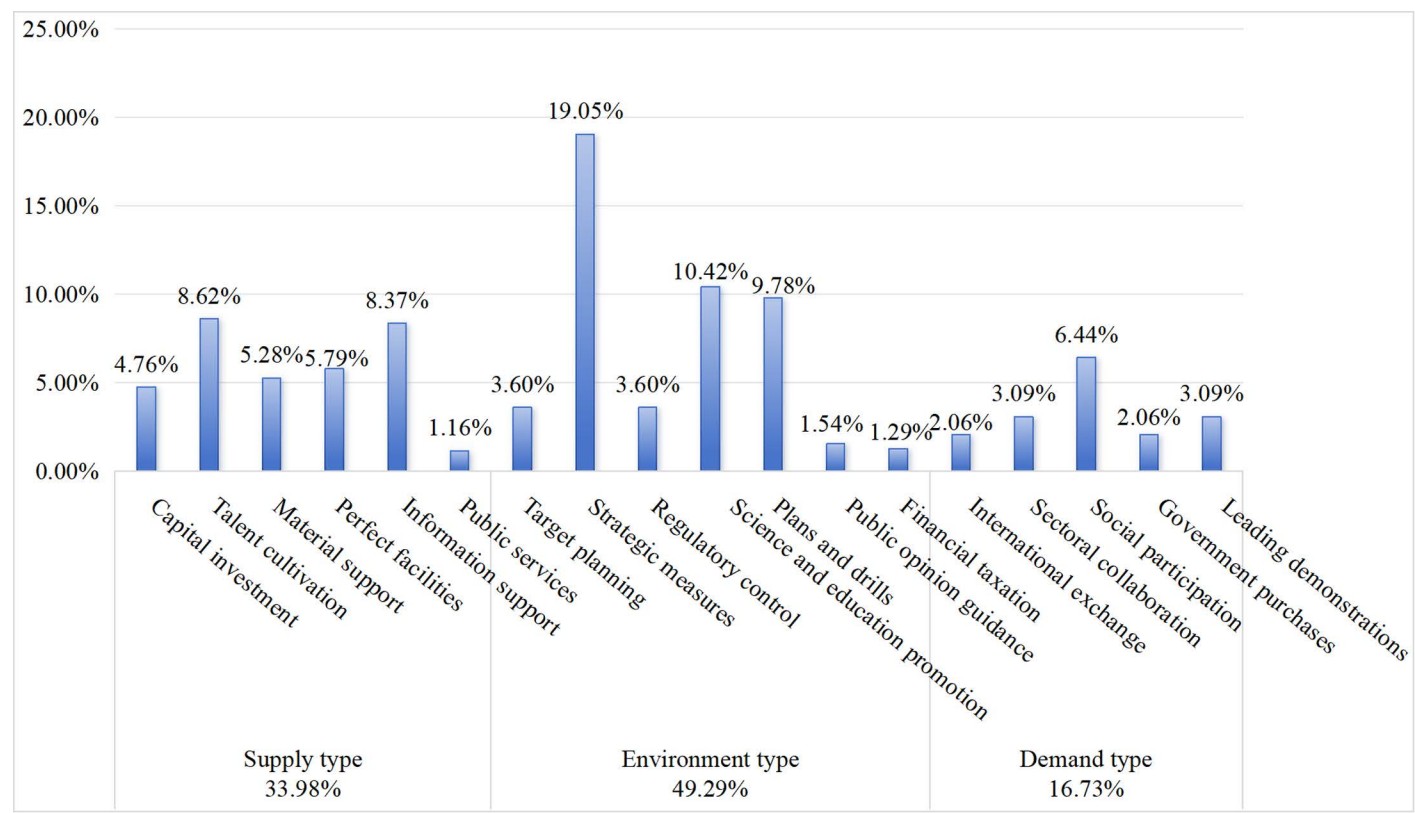

**Fig 5. Proportion of policy tools utilized.**

emergency management is important to the Chinese government, which is different from general community services and requires a higher level of professionalism. Hence, emphasis on training professionals and teams will help organizations organize effective relief immediately when community emergencies occur and reduce economic losses and casualties caused by disasters promptly. Second, the ratio of "information support" is 8.37%. On the one hand, strengthening the construction of emergency response platforms, further optimizing the collection and processing of risk information and analyzing through the establishment of information platforms and networks are important to the Chinese government; on the other hand, the government places considerable value on the research and application of public security technologies, as well as using high technology, such as the Internet of Things and Beidou navigation, to improve risk prevention capabilities. Additionally, the use of "perfect facilities", "material support" and "capital investment" is relatively balanced, accounting for 5.79%, 5.28% and 4.76%, respectively. However, the frequency of using "public services" policy tools is low, at only 1.16%, which reflects that the government has not paid enough attention to using various measures to promote public services such as consulting services and residents' care services. Therefore, we should promote the internal balance of supply-type policy tools and further increase investing in funds, materials and public services to better develop community emergency management.

With respect to environmental-type policy tools, the Chinese government has influenced community emergency management through "strategic measures" and "science and education promotion". The strategic measurement tools provide grassroots communities with specific response measures, procedures and principles for emergency management, improving

response speed and standardizing response activities. Promoting science and education is manifested mainly in the Chinese government's efforts to increase community residents' awareness and ability to prevent disasters by publicizing emergency knowledge through text messages and radio broadcasts, as well as providing education and training to residents. However, the use of policy tools, namely, "public opinion guidance" and "financial taxation", is relatively infrequent, accounting for only 1.54% and 1.29% of the total, respectively, leaving a significant gap to be addressed. The government's efforts to guide public opinion have been helpful in grasping the correct direction of public opinion and continuously improving the level of news reporting, thus enhancing the confidence of community residents in the fight against the disaster. Moreover, the government's financial support through loan subsidies and tax incentives is conducive to mobilizing and giving full play to the enthusiasm and role of enterprises and forming a pattern of diversified participation by social forces. Therefore, we should further strengthen the use of "public opinion guidance" and "financial taxation" [39] to continuously realize relevant policy objectives of community emergency management.

Among the demand-type policy tools, the Chinese government is most inclined to use "social participation", which aims to mobilize the public to participate in emergency management activities and transform the government's single-dominant model into a multigovernance model. However, as shown in Fig 5, compared with the other two policy tools, demand-type policy tools account for the smallest proportion of the three categories. Furthermore, its most frequently used subcategory, "social participation", has a coverage rate of only 6.44% of all policy tools, which is relatively low. Moreover, "international exchange" and "government purchases" policy tools are used less frequently, accounting for 2.06%. To a certain extent, the underutilization of the above two types of demand-type policies has blocked the stimulating effect of international exchanges and government purchases on the demand for community emergency management. Therefore, on the one hand, it is necessary to strengthen leading demonstrations and sectoral collaboration, promote point-to-area development by establishing demonstration communities and industries, and, at the same time, promote dynamic cooperation among government departments to give full play to their comparative advantages. On the other hand, it is necessary to pay more attention to international exchanges and government purchases, actively learn from excellent foreign emergency treatment methods and practical research, and encourage relevant departments to purchase emergency products and services from the market to improve the efficiency of emergency rescue.

### 5.4 Dimensional analysis of policy objectives

The categorization statistics of the policy objective dimensions are shown in Fig 6. Overall, the subcategories of policy objectives involved in China's CEMPs are, in order of importance, 32.95% to improve the level of emergency management, 28.19% to improve the efficiency of emergency rescue, 27.03% to strengthen community resilience to disasters, and 11.84% to improve residents' emergency literacy. The results indicate that the Chinese government's reform tasks related to community emergency management have different degrees of urgency.

First, it is committed to measures such as "sounding regulations and preplanning" (10.30%), "improving institutional mechanisms" (8.37%), and "strengthening supervision and management" (7.85%) to promote the improvement of community emergency management. However, implementing treatment rewards must be further strengthened. Emergency management is an important function of the government to ensure the safety of people's lives and property. Improving the enthusiasm of relevant staff to implement the goal of refining the treatment and reward of grassroots communities and evaluating the monitoring standards is beneficial.

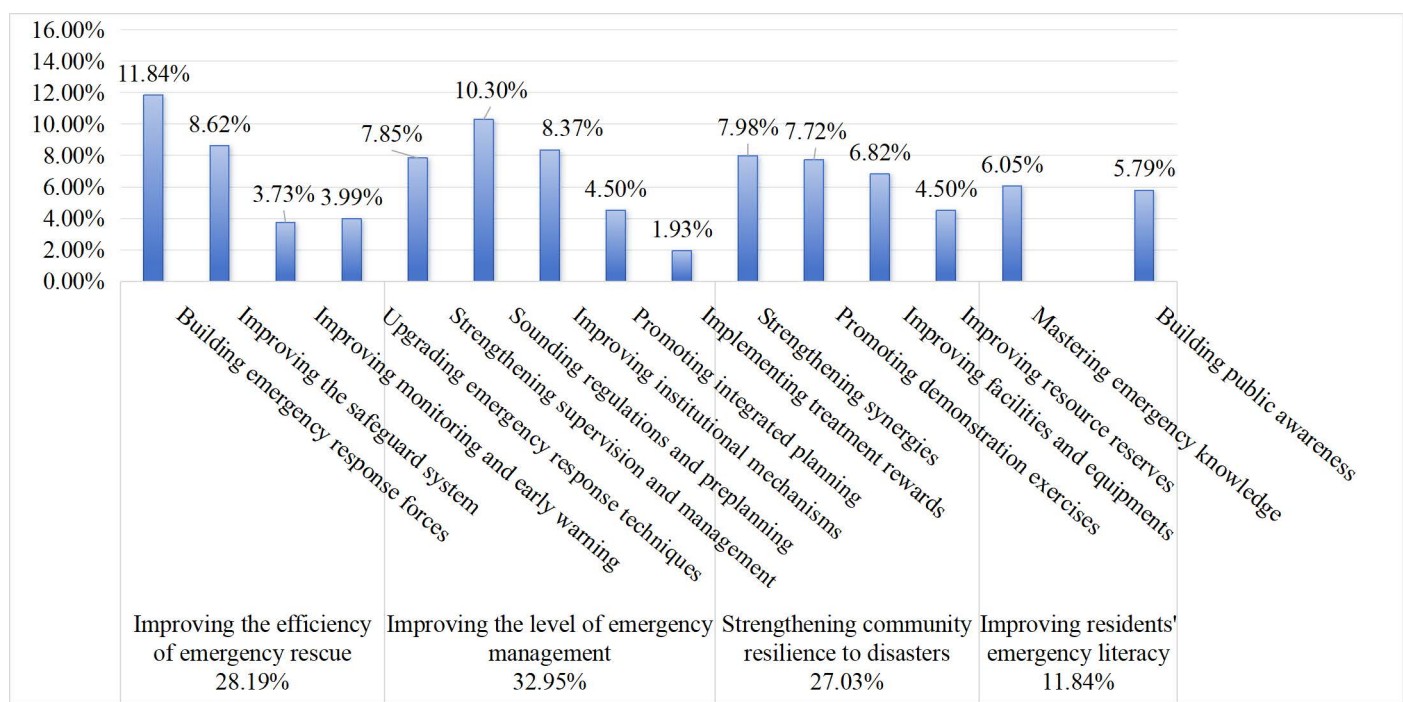

**Fig 6. Proportion of policy objectives utilized.**

Second, the two target dimensions of improving the efficiency of emergency rescue and strengthening community resilience to disasters have also gained increasing attention in recent years. The government aims to improve the community's ability to cope with crises through "building emergency response forces" (11.84%) and "strengthening synergies" (7.98%), whereas the policy objectives of "improving monitoring and early warning" and "improving resource reserves" are slightly less well planned. Therefore, efforts should be made to improve emergency monitoring technology and emergency resource reserves to solve problems such as the limitation of rescue technology and the shortage of emergency resources, among which a sound resource allocation mechanism is particularly important to ensure the orderly allocation of resources [40].

Finally, the policy objectives are superior in terms of improving residents' emergency literacy, and the percentages of "mastering emergency knowledge" and "building public awareness" are close to each other, at only 6.05% and 5.79%, respectively. Community emergency response is a universal endeavor that requires not only support from the government and assistance from the market but also the participation of the community residents. Therefore, we should clarify the core position of knowledge in community emergency management [41], build a "normalization+long-term" mechanism of emergency training, and incorporate emergency training into national basic education to continuously improve community residents' emergency knowledge and skills.

### 5.5 Analysis of the crosscutting dimension of policy tools–policy objectives

The distribution of the subcategories of the community emergency response policy objectives among the policy tools in China is shown in Fig 7. In terms of improving the efficiency of emergency rescue, the Chinese government most often uses supply-type policy tools (70.78%), with "talent cultivation" (28.77%) and "information support" (21.00%) as the principal methods.

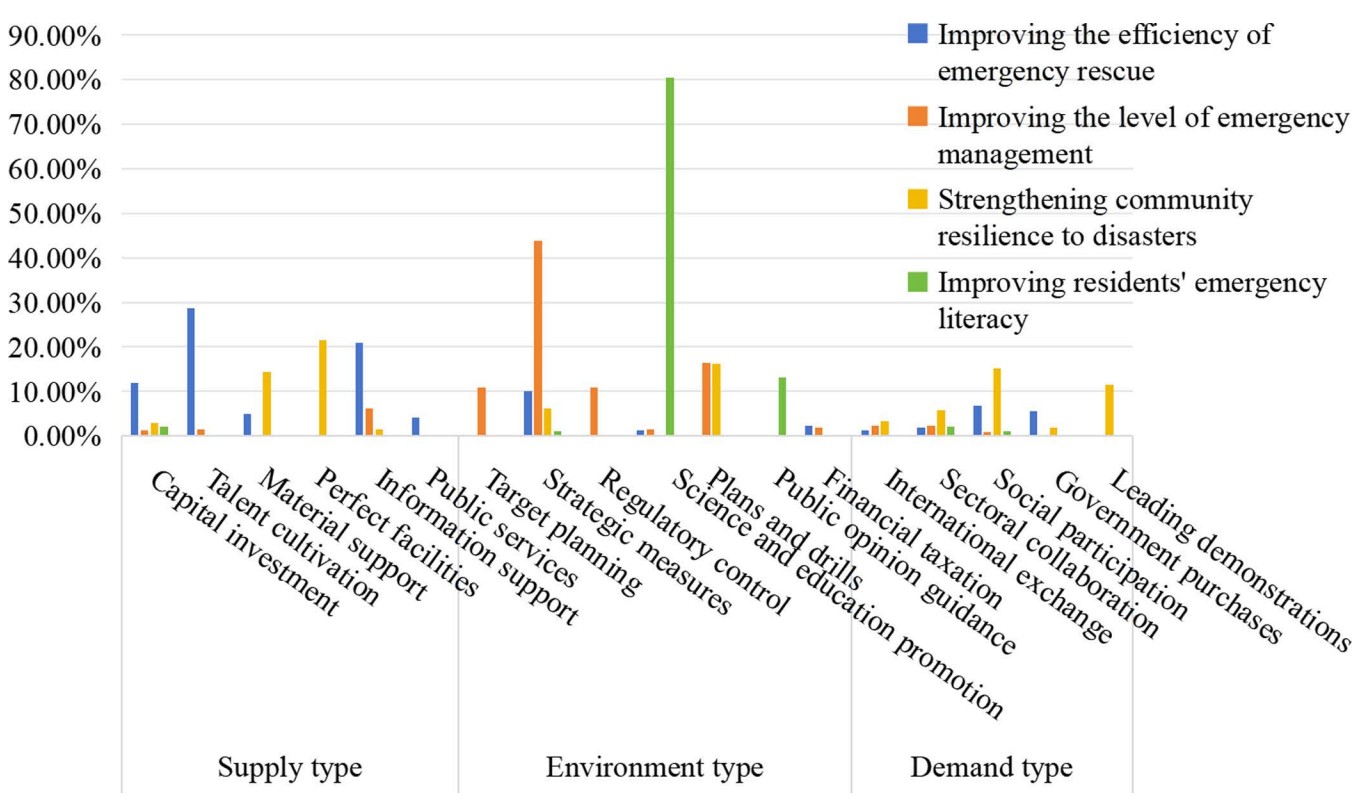

**Fig 7. Distribution statistics of policy objective subcategory dimensions in policy tools.**

The former focuses on cultivating professional talent and expert teams for community emergency management; the latter is committed to improving the construction of information platforms and providing information support for scientific decision-making in community emergency management. The second is demand-type policy tools (15.53%), of which "social participation" accounts for the largest proportion (6.85%). The reason is that disasters are sudden and accidental. By encouraging community residents, enterprises and social organizations to actively participate in the daily life of community emergency management, we can give full play to the market and social demand in improving the efficiency of community emergency rescue and reducing disaster losses in time. Finally, there are environmental-type policy tools (13.70%), of which "strategic measures" account for the greatest proportion (10.05%), which shows that the community tends to promote the realization of policy objectives through supervision and evaluation, personnel incentives and assessment, accountability and other measures. In the future, policy optimization can adopt environmental-type policy tools represented by strategic measures and enrich the use of environmental and demand tools. Policy optimization should also play the important role of "government–market–society" in improving the effectiveness of community emergency rescue.

In terms of improving the level of emergency management, the Chinese government tends to use environmental-type policy tools (85.55%), especially "strategic measures" (43.75%) and "plans and drills" (16.41%), more frequently than other subcategories. Therefore, goal-oriented performance appraisal, evaluation and monitoring, emergency plans, emergency drills and other measures are the preferred ways for the Chinese government to improve the level of community emergency management, whereas supply-type policy tools, such as the construction of community emergency response infrastructure and public services, account

for almost zero. Moreover, demand-type policy tools, such as "social participation", "government purchases" and "leading demonstrations", still must be further strengthened. Therefore, we should appropriately increase the proportion of supply-type policy tools, such as "perfect facilities" and "public services", as well as demand-type policy tools, such as "social participation" and "government purchases", to internally promote the government and externally promote the market.

To strengthen community resilience to disasters, all three types of policy tools are used to a certain extent: supply-type policy tools account for 40.00%, demand-type policy tools account for 37.62%, and environment-type policy tools account for 22.38%. The former is dominated by "perfect facilities" (21.43%) and "material support" (14.29%), indicating that the Chinese government tends to improve the hardware supply for emergency response and rescue through infrastructure construction and emergency material protection, thus enhancing the resilience of grassroots communities to disasters. These two supply-type policy tools are often combined with other policy tool subcategories, mainly because the supply of emergency hardware usually needs to be accomplished in conjunction with funding, information, services, and markets. Among the latter two, the frequencies of "social participation" (15.24%) and "plans and drills" (16.19%) are similar, which shows that the active participation of the public is indispensable to ensure the smooth implementation of emergency drills. The correlation between environmental-type policy tools and this goal has not been fully explored; thus, we should actively explore the potential matching of environmental-type policy tools and rationally allocate policy resources.

In terms of improving residents' emergency literacy, environmental-type policy tools (94.57%) far exceeded the other two categories. "science and education promotion" (80.43%) and "public opinion guidance" (13.04%) are among the top two policy tool subcategories. This suggests that the Chinese government prefers to increase the emergency response awareness and capacity of community residents by providing them with science education and publicity activities related to emergency management, as well as creating a favorable public opinion environment for emergencies. Therefore, it is necessary to balance and optimize the combination of the three types of policy tools, focusing on the role of supply- and demand-type tools in realizing this policy goal.

## 6. Research suggestions

In this paper, 87 CEMPs (777 policy rules) promulgated in China from 2004 to 2024 are taken as research samples. Content analysis and social network analysis are used to systematically sort out and quantitatively evaluate the CEMPs in China by constructing a two-dimensional analysis framework of "policy tools–policy objectives". Some problems in China's CEMPs, such as the uneven use of policy tools, the unbalanced internal structure of policy tools, the large distribution gap of policy target elements, and the adaptability between policy tools and policy target elements, must be improved. To this end, the following policy suggestions are proposed: 1) Appropriately increase the frequency of using demand-type policy tools: Given that the "supply-type+environment-type" policy tools model does not sufficiently stimulate domestic demand, it is necessary to further increase decentralization, strengthen the cross-border exchange of community emergency management and its achievements, actively promote the formation of a collaborative governance model of community emergency management with "government-led, departmental cooperation and community participation" [42], scientifically evaluate the multiparty collaborative effect, and continuously optimize the collaborative governance model according to the evaluation results. Additionally, when evaluating the effectiveness of the collaborative governance model of community emergency management, we should focus on the characteristics of public emergencies and strive to promote

a standardized, orderly, scientific, and efficient multiparty collaborative governance model. 2) Effectively optimize the internal structure of the three types of policy tools: First, the use of supply-type policy tools such as funds and public services should be increased. Second, for environmental-type policy tools, the top-level design of the emergency management system should be optimized, and the advantages of comprehensive planning should be fully exploited. Finally, for demand-type policy tools, attention should be given to "international exchange" and "government purchases". 3) Comprehensively deepen the reform of CEMP objectives: We should strengthen the implementation of personnel treatment and rewards, improve monitoring and early warning, improve emergency resource reserves and establish community residents' public awareness of emergencies to realize the four types of policy objectives. 4) Reasonably improve the adaptability between policy tools and policy objectives: The selection and allocation of policy tools should be combined with the significance and principles of CEMP objectives [43] so that they can be tailored to local conditions and accurately oriented to fully release the cross-effects of policies.

## 7. Conclusion

CEMPs play a crucial role in regulating and promoting the construction of grassroots community emergency management. This paper constructs a two-dimensional analytical framework of "policy tools-policy objectives", uses content analysis and social network analysis methods to quantitatively research on China's CEMP texts, summarizes and analyzes the types of policy tools and their drawbacks, and proposes targeted policy recommendations. The research results not only fill the theoretical gap in the field of emergency management at the community level, but also provide strong basis for future policy formulation and optimization, which will help further improve the community emergency management system and promote the enhancement of community emergency management capabilities.

Specifically, first, this paper takes policy tools as a theoretical perspective, and based on the content of China's CEMPs, adopts Rothwell and other policy tool classification standards to reasonably classify policy tools suitable for this field, dividing them into three types of policies: demand-type, supply-type, and environmental-type, and including 6, 7, and 5 sub policy types respectively. The research not only enriches the field of policy tool theory, but also opens up new research paths to reveal the problems existing in current CEMPs. Second, based on the content of the CEMP texts and the characteristics of Chinese community emergency management, the objectives of the CEMPs were creatively deconstructed. This paper divides the policy objectives into four aspects: improving the efficiency of emergency rescue, improving the level of emergency management, strengthening community resilience to disasters, and improving residents' emergency literacy. The division of this goal has high theoretical value and practical significance, providing core evaluation criteria for the planning and implementation of CEMP plans. Third, this paper constructs a two-dimensional analysis framework for China's CEMPs from two dimensions: policy tools and policy objectives. The proposal of this analysis framework injects new vitality into the quantitative evaluation of policies. Finally, in response to the shortcomings of China's CEMPs, such as the uneven use of policy tools, the unbalanced internal structure of policy tools, the large distribution gap of policy objective elements, and the adaptability between policy tools and policy objectives, this paper proposes targeted and actionable improvement suggestions after in-depth analysis, providing a reasonable optimization path for improving China's CEMP system.

However, this study has the following limitations: ① Due to the large time span and number of samples, only policies at the central level were selected without considering the differences between central- and local-level policies, and a comparative analysis of policies at different levels has not yet been conducted. ② The environmental differences between

localities have not been discussed, and a more detailed study of local policies in natural disaster-prone areas, such as earthquakes, floods, and tsunamis, as well as in densely populated areas, has not yet been conducted. In future research, using a larger sample, we will investigate the characteristics of regional combinations of disaster occurrences, compare the differences between central and local levels, and further explore the laws of developing community emergency management policies with both Chinese and regional characteristics.

## Author contributions

**Conceptualization:** Hehua Du.

**Data curation:** Xingyue Wang.

**Investigation:** Xingyue Wang.

**Project administration:** Hehua Du.

**Supervision:** Hehua Du.

**Writing – original draft:** Hehua Du.

**Writing – review & editing:** Hehua Du.

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
