## [Decision Letter · Decision Letter 0]

16 Oct 2024

PONE-D-24-12158Research on Community Emergency Management Policy in China Based on Policy Text ToolPLOS ONE

Dear Dr. Du,

Thank you for submitting your manuscript to PLOS ONE. After careful consideration, we feel that it has merit but does not fully meet PLOS ONE’s publication criteria as it currently stands. Therefore, we invite you to submit a revised version of the manuscript that addresses the points raised during the review process.lease submit your revised manuscript by Nov 30 2024 11:59PM. If you will need more time than this to complete your revisions, please reply to this message or contact the journal office at plosone@plos.org . Please include the following items when submitting your revised manuscript:

We look forward to receiving your revised manuscript.

Kind regards,

Chenfeng Xiong

Academic Editor

PLOS ONE

Journal Requirements:

2. We suggest you thoroughly copyedit your manuscript for language usage, spelling, and grammar. If you do not know anyone who can help you do this, you may wish to consider employing a professional scientific editing service. The American Journal Experts (AJE) (https://www.aje.com/) is one such service that has extensive experience helping authors meet PLOS guidelines and can provide language editing, translation, manuscript formatting, and figure formatting to ensure your manuscript meets our submission guidelines. Please note that having the manuscript copyedited by AJE or any other editing services does not guarantee selection for peer review or acceptance for publication. Upon resubmission, please provide the following: ● The name of the colleague or the details of the professional service that edited your manuscript ● A copy of your manuscript showing your changes by either highlighting them or using track changes (uploaded as a *supporting information* file) ● A clean copy of the edited manuscript (uploaded as the new *manuscript* file)

4. Thank you for stating the following financial disclosure: “This research is supported by the project of "Wisdom Emergency Management Key Laboratory" (No. 2023ZHYJGL-4) of the Sichuan Provincial Key Laboratory of Philosophy and Social Sciences and the Youth Fund for Humanities and Social Sciences Research of the Ministry of Education of the People's Republic of China (No. 23YJCZH051).”

5. We note that your Data Availability Statement is currently as follows: “All relevant data are within the manuscript and in Supporting Information files.”

Please confirm at this time whether or not your submission contains all raw data required to replicate the results of your study. Authors must share the “minimal data set” for their submission. PLOS defines the minimal data set to consist of the data required to replicate all study findings reported in the article, as well as related metadata and methods (https://journals.plos.org/plosone/s/data-availability#loc-minimal-data-set-definition). For example, authors should submit the following data: - The values behind the means, standard deviations and other measures reported; - The values used to build graphs; - The points extracted from images for analysis. Authors do not need to submit their entire data set if only a portion of the data was used in the reported study. If your submission does not contain these data, please either upload them as Supporting Information files or deposit them to a stable, public repository and provide us with the relevant URLs, DOIs, or accession numbers. For a list of recommended repositories, please see https://journals.plos.org/plosone/s/recommended-repositories. If there are ethical or legal restrictions on sharing a de-identified data set, please explain them in detail (e.g., data contain potentially sensitive information, data are owned by a third-party organization, etc.) and who has imposed them (e.g., an ethics committee). Please also provide contact information for a data access committee, ethics committee, or other institutional body to which data requests may be sent. If data are owned by a third party, please indicate how others may request data access.

Additional Editor Comments:

The editor agrees with the two reviewers that the paper does have its scientific merit in advancing the understanding of emergency management policy in China. Yet, the paper needs a thorough revision to increase its quality and paper organization before it can be accepted for publication at PLOS One.

In addition to the reviewers' comments, the Editor has the following major comments:

First of all, the paper needs to have a much better discussion of their methodology. There are multiple methodlogical components of the study, including text selection, content analysis, (social?) network analysis, policy analysis, etc. It is warranted that the authors create an overarching framework (or a flowchart at the higher level) that integrates these parts together (along with input data, and outputs/policy recommendations). The paper in its current form is very fragmented and needs a methodological framework as its backbone.

Secondly, please make thorough check on the methodology description and make sure they are all adequately discussed. For instance, the content analysis and (social?) network analysis need more discussion and technical details. Also, arguably the method is no longer a social network problem but rather you used network and graph to represent different linkages of texts/words, if the Editor understood it correctly. Then, such a generalization needs to be formulated and discussed in the paper.

Third, please revise the policy recommendation and embed them in your Section 5. This is because many of your recommendations are backed up by your analysis findings, they should be discussed inline with your anlaysis results/visualizations. Evidences should then be cited to support each policy implication/recommendation item. In Section 6, you merely revisit these recommendations.

Last but not least, the paper lacks a discussion of contribution. Please highlight your innovation, whether it is within technical/methodological frame, or about empirical findings/policy suggestions, the readers should be informed on why this article is cutting-edge and has the value for publication at an impactful journal such as PLOS One.

Reviewers' comments:

Reviewer's Responses to Questions

**Comments to the Author**

1. Is the manuscript technically sound, and do the data support the conclusions?

Reviewer #1: Yes

Reviewer #2: Yes

2. Has the statistical analysis been performed appropriately and rigorously? 

Reviewer #1: Yes

Reviewer #2: Yes

3. Have the authors made all data underlying the findings in their manuscript fully available?

Reviewer #1: Yes

Reviewer #2: Yes

4. Is the manuscript presented in an intelligible fashion and written in standard English?

Reviewer #1: No

Reviewer #2: Yes

5. Review Comments to the Author

Reviewer #1: The research of community emergency management in China is a field worthy of attention, and the selection of this field is worthy of full affirmation. Based on the analysis of policy texts, this paper puts forward reasonable suggestions for community emergency management in China, which is also worthy of affirmation. On the whole, the article is worth publishing. However, it is also necessary to pay attention to the following problems: First, there are too few policy text names listed in Table 1, and the policy text names directly related to the topic of this paper should be listed. The number of existing policy text names is small, and it is not clear which policy text is directly related to this paper. Second, there are many errors in the language of this paper. It is recommended that professional service organizations make comprehensive changes to the text language.

Reviewer #2: I reviewed the paper titled “Research on Community Emergency Management Policy in China Based on Policy Text Tool.” The author analyzed the current status and existing problems of the CEMP system and provided suggestions for the usage of policy instruments, reforming policy goals, and assessing the degree of compatibility.

I have several comments and questions:

Abstract: "community emergency management policy (CEMP)" should be changed to "Community Emergency Management Policy (CEMP)" for consistency.

Line 13: There is an inconsistency between "two dimensional analysis framework of 'policy tool-policy goal'" and line 128: "two-dimensional analysis framework of 'policy tools-policy objectives'." These terms should be unified.

Line 143: The word "plans" is duplicated.

Line 482: In the 6.2 Policy Recommendations section, I suggest adding steps for emergency management evaluation. Based on the evaluation results, identify the effectiveness of policy tools and then give recommendations according to their effectiveness. The emergency scenario should be considered when evaluating the effectiveness of policy tools.

Questions:

Q1: On Line 140, would using "community emergency" as a keyword provide more comprehensive data results?

Q2: On Line 141, why did you choose those three keywords? Several other keywords, such as "emergency response," "emergency rescue," and "emergency drills," are mentioned frequently in the paper. Why were they not selected?

Q3: On Line 216, are the tools listed in Table 2, "Division of CEMP Tools," sourced from the Magic Weapon Legal Database of Peking University?

6. PLOS authors have the option to publish the peer review history of their article (what does this mean? ). If published, this will include your full peer review and any attached files.

**Do you want your identity to be public for this peer review?** For information about this choice, including consent withdrawal, please see our Privacy Policy .

Reviewer #1: **Yes: ** bangfan liu

Reviewer #2: **Yes: ** Ya Ji

---

## [Author Response · Author response to Decision Letter 0]

1 Dec 2024

We sincerely thank the editor and all reviewers for their professional comments on our articles, which helped us improve our manuscript. The point-by-point response to the comments is given below.

Comments from the editor：

General Comments:

The editor agrees with the two reviewers that the paper does have its scientific merit in advancing the understanding of emergency management policy in China. Yet, the paper needs a thorough revision to increase its quality and paper organization before it can be accepted for publication at PLOS One.

Specific comments:

1.First of all, the paper needs to have a much better discussion of their methodology. There are multiple methodlogical components of the study, including text selection, content analysis, (social?) network analysis, policy analysis, etc. It is warranted that the authors create an overarching framework (or a flowchart at the higher level) that integrates these parts together (along with input data, and outputs/policy recommendations). The paper in its current form is very fragmented and needs a methodological framework as its backbone.

Answer: We agree. We think this is a good suggestion. By building an overall framework to integrate the methods used, the context of the article can be clearer. Therefore, we have created a general research framework, and put it in Part 3.1 of this paper, and added a frame diagram (see Fig. 1)(Line. 132-148).

2.Secondly, please make thorough check on the methodology description and make sure they are all adequately discussed. For instance, the content analysis and (social?) network analysis need more discussion and technical details. Also, arguably the method is no longer a social network problem but rather you used network and graph to represent different linkages of texts/words, if the Editor understood it correctly. Then, such a generalization needs to be formulated and discussed in the paper.

Answer: We agree. We have thoroughly checked the method description according to the editor's suggestion, and added more discussion and technical details to the research method in part 3.3 of the paper, and added an exposition of the use of network and graphic to express the relationship between keywords in social network analysis. (Line.176-179; 186-194; 312-317)

3.Third, please revise the policy recommendation and embed them in your Section 5. This is because many of your recommendations are backed up by your analysis findings, they should be discussed inline with your anlaysis results/visualizations. Evidences should then be cited to support each policy implication/recommendation item. In Section 6, you merely revisit these recommendations.

Answer: We agree. We have revised the policy recommendations according to the editor's suggestions and embedded them in Part 5, discussed them together with the research results, and added references to supporting evidence, and revisited these policy recommendations in Part 6. (Line.332-344; 366-368; 386-388; 400-408; 423-427; 434-438; 444-448; 470-474; 484-488; 502-505; 513-515; 519-552)

4.Last but not least, the paper lacks a discussion of contribution. Please highlight your innovation, whether it is within technical/methodological frame, or about empirical findings/policy suggestions, the readers should be informed on why this article is cutting-edge and has the value for publication at an impactful journal such as PLOS One.

Answer: We agree. We have emphasized our innovation and the value of the article in the research framework and policy suggestions. (Line.133-136; 554-557)

Comments from the reviewers:

Reviewer 1

General Comments:

The research of community emergency management in China is a field worthy of attention, and the selection of this field is worthy of full affirmation. Based on the analysis of policy texts, this paper puts forward reasonable suggestions for community emergency management in China, which is also worthy of affirmation. On the whole, the article is worth publishing. However, it is also necessary to pay attention to the following problems:

Specific comments:

1. First, there are too few policy text names listed in Table 1, and the policy text names directly related to the topic of this paper should be listed. The number of existing policy text names is small, and it is not clear which policy text is directly related to this paper.

Answer: We agree. We sincerely thank you for your valuable suggestions. We have added policy texts directly related to the theme of this article in Table 1, which makes the display of policy texts clearer. Such as “Opinions on Further Playing the Role of Emergency Broadcasting in Emergency Management”; “Opinions on Improving and Perfecting the Comprehensive Service Functions at the Village Level” , etc.(Line.166)

2.Second, there are many errors in the language of this paper. It is recommended that professional service organizations make comprehensive changes to the text language.

Answer: We agree. Thank you for your advice. We found a professional organization (AJE) to polish our article, and we tried our best to revise it. These changes will not affect the content and framework of the paper, so we didn't list the changes here. In addition, according to the suggestion of AJE, we revised the title of the article to “Community Emergency Management Policy in China Using a Policy Text Tool”. We uploaded the polished version to the supporting information, and we hope that the revised manuscript will be accepted by you. (Line.1-2)

Reviewer 2

1.Abstract: "community emergency management policy (CEMP)" should be changed to "Community Emergency Management Policy (CEMP)" for consistency.

Answer: We agree. We have made corresponding changes in the abstract to maintain consistency. (Line. 11)

2.Line 13: There is an inconsistency between "two dimensional analysis framework of 'policy tool-policy goal'" and line 128: "two-dimensional analysis framework of 'policy tools-policy objectives'." These terms should be unified.

Answer: We are very sorry for our careless mistakes. Thank you for reminding us. We have unified the terms in these two places. (Line.15 and Line126)

3.Line 143: The word "plans" is duplicated.

Answer: We are very sorry for our careless mistakes. We have deleted the redundant words.(Line.154)

4.Line 482: In the 6.2 Policy Recommendations section, I suggest adding steps for emergency management evaluation. Based on the evaluation results, identify the effectiveness of policy tools and then give recommendations according to their effectiveness. The emergency scenario should be considered when evaluating the effectiveness of policy tools.

Answer: Thank you for your suggestion. We have added relevant explanations to the assessment in the emergency management assessment step, and suggested that emergency scenarios should be considered. (lines 533-538) .

5.On Line 140, would using "community emergency" as a keyword provide more comprehensive data results?

Answer: We sincerely thank you for your careful reading. We tried to use "community emergency" as the key word to search. In addition to the policy texts already covered by other key words, we supplemented the policy text "Opinions on Further Improving the Emergency Management Capability at the Grass-Roots Level" (2024) and listed it in Table 1. (Line.158 and Line.166)

6.On Line 141, why did you choose those three keywords? Several other keywords, such as "emergency response," "emergency rescue," and "emergency drills," are mentioned frequently in the paper. Why were they not selected?

Answer: We sincerely thank you for your careful reading. The keywords we selected cover the policy texts about community emergency management, but we still try to search carefully again with keywords such as "emergency response", "community emergency rescue" and "community emergency drill", and supplement the policy texts by adding five policy texts such as "People's Republic of China (PRC) Emergency Response Law" (2024). As shown in Table 1. (158-159; 166)

In addition, we extended the policy text retrieval time from 2004-2023 to 2004-2024. At the same time, after adding relevant search keywords, the number of policy texts increased from 81 to 87. "5.Quantitative analysis of CEMP Texts" has updated the corresponding charts and data in the text.(See Fig3; Fig5; Fig6; Fig7 ) (Line.272; 324-327; 348; 354; 363-364; 379; 395-397; 414-416; 420-421; 431-432; 441; 455-456; 461; 466-467; 476-477; 490-493; 500;507-508)

7. On Line 216, are the tools listed in Table 2, "Division of CEMP Tools," sourced from the Magic Weapon Legal Database of Peking University?

Answer: Clarification. Thank you for your question. The tools classification listed in Table 2 "Division of CEMP Tools" adopts Rothwell and other policy tools classification standards. See Reference [34] for details.(Line.211-213; 223-225)

---

## [Decision Letter · Decision Letter 1]

1 Jan 2025

PONE-D-24-12158R1Community Emergency Management Policy in China Using a Policy Text ToolPLOS ONE

Dear Dr. Du,

Thank you for submitting your manuscript to PLOS ONE. After careful consideration, we feel that it has merit but does not fully meet PLOS ONE’s publication criteria as it currently stands. Therefore, we invite you to submit a revised version of the manuscript that addresses the points raised during the review process.

 Thank you for revising your manuscript according to editor's and reviewers' comments. The two reviewers are happy with the revision. However, the editor's original comments are not fully addressed. First of all, the authors' responses provided page and line number of the revision to each comment, however, the line number seemed not exactly accurate. Can you double check and make sure the pointed location is exactly what you have revised for each of the comments? Then, specifically, the authors did not address the contribution concern well, which was my fourth point in the Round-1 review. What exactly is innovative in this research? What contributions made by the article warrant a publication in Plos-One? Please explain explicitly in the paper's conclusion section. The current responses from the authors have pointed their "contributions" to the paper's "Limitation" section, which is confusing. 

We look forward to receiving your revised manuscript.

Kind regards,

Chenfeng Xiong

Academic Editor

PLOS ONE

Journal Requirements:

Reviewers' comments:

Reviewer's Responses to Questions

**Comments to the Author**

1. If the authors have adequately addressed your comments raised in a previous round of review and you feel that this manuscript is now acceptable for publication, you may indicate that here to bypass the “Comments to the Author” section, enter your conflict of interest statement in the “Confidential to Editor” section, and submit your "Accept" recommendation.

Reviewer #1: (No Response)

Reviewer #2: All comments have been addressed

2. Is the manuscript technically sound, and do the data support the conclusions?

Reviewer #1: Yes

Reviewer #2: Yes

3. Has the statistical analysis been performed appropriately and rigorously? 

Reviewer #1: Yes

Reviewer #2: Yes

4. Have the authors made all data underlying the findings in their manuscript fully available?

Reviewer #1: Yes

Reviewer #2: Yes

5. Is the manuscript presented in an intelligible fashion and written in standard English?

Reviewer #1: Yes

Reviewer #2: Yes

6. Review Comments to the Author

Reviewer #1: In response to the review comments, the author of the article has carefully considered and made corresponding modifications or responses. The revised manuscript has met the publication standards and it is recommended to publish it.

Reviewer #2: Thanks! I have no other comments on this paper, my previous comments are addresses and answsered clearly.

7. PLOS authors have the option to publish the peer review history of their article (what does this mean? ). If published, this will include your full peer review and any attached files.

**Do you want your identity to be public for this peer review?** For information about this choice, including consent withdrawal, please see our Privacy Policy .

Reviewer #1: **Yes: ** liu bangfan

Reviewer #2: **Yes: ** Ya Ji

---

## [Author Response · Author response to Decision Letter 1]

5 Jan 2025

We sincerely thank the editor for his professional comments on our articles, which helps us to improve our manuscripts. The point-by-point response to the comments is given below.

Comments from the editor：

General Comments:

Thank you for revising your manuscript according to editor's and reviewers' comments. The two reviewers are happy with the revision. However, the editor's original comments are not fully addressed.

Specific comments:

1.First of all, the authors' responses provided page and line number of the revision to each comment, however, the line number seemed not exactly accurate. Can you double check and make sure the pointed location is exactly what you have revised for each of the comments?

Answer: Thank you for your careful inspection. We are sorry for our carelessness. According to your opinion, we made corrections so that the revised page number and line number of each comment in reply correspond to the position in the text one by one. See "Correction" below for the specific modifications, and the modified contents have been marked in red. (See the second page of this letter for details.)

2.Then, specifically, the authors did not address the contribution concern well, which was my fourth point in the Round-1 review. What exactly is innovative in this research? What contributions made by the article warrant a publication in Plos-One? Please explain explicitly in the paper's conclusion section. The current responses from the authors have pointed their "contributions" to the paper's "Limitation" section, which is confusing.

Answer: Thank you for your professional advice. We have made a serious revision according to your suggestion. In the conclusion of chapter 7, we clearly explain the innovation and contribution value of this study.(Line. 553-586)

Correction:

Comments from the editor：

4.Last but not least, the paper lacks a discussion of contribution. Please highlight your innovation, whether it is within technical/methodological frame, or about empirical findings/policy suggestions, the readers should be informed on why this article is cutting-edge and has the value for publication at an impactful journal such as PLOS One.

Answer: We agree. We have emphasized our innovation and the value of the article in the research framework and policy suggestions. (Line.133-136; 553-586)

Comments from the reviewers:

Reviewer 2

6.On Line 141, why did you choose those three keywords? Several other keywords, such as "emergency response," "emergency rescue," and "emergency drills," are mentioned frequently in the paper. Why were they not selected?

Answer: We sincerely thank you for your careful reading. The keywords we selected cover the policy texts about community emergency management, but we still try to search carefully again with keywords such as "emergency response", "community emergency rescue" and "community emergency drill", and supplement the policy texts by adding five policy texts such as "People's Republic of China (PRC) Emergency Response Law" (2024). As shown in Table 1. (158-159; 166)

In addition, we extended the policy text retrieval time from 2004-2023 to 2004-2024. At the same time, after adding relevant search keywords, the number of policy texts increased from 81 to 87. "5.Quantitative analysis of CEMP Texts" has updated the corresponding charts and data in the text.(See Fig3; Fig5; Fig6; Fig7 ) (Line.17; 163-164; 259; 272; 297; 324; 326; 327; 348; 354; 362-364; 379; 395; 397; 410; 414-416; 420-421; 431-432; 441; 450; 455-456; 461; 466-467; 476-477; 490-493; 500; 507-508; 517)

---

## [Editor Report · Decision Letter 2]

30 Jan 2025

Community Emergency Management Policy in China Using a Policy Text Tool

PONE-D-24-12158R2

Dear Dr. Du,

We’re pleased to inform you that your manuscript has been judged scientifically suitable for publication and will be formally accepted for publication once it meets all outstanding technical requirements.

Kind regards,

Chenfeng Xiong

Academic Editor

PLOS ONE

Additional Editor Comments (optional):

Thank you for the effort addressing my remaining comments. The updated version reads good to me. Thumb up.
---

## [Editor Report · Acceptance letter]

PONE-D-24-12158R2

PLOS ONE

Dear Dr. Du,

I'm pleased to inform you that your manuscript has been deemed suitable for publication in PLOS ONE. Congratulations! Your manuscript is now being handed over to our production team.

Kind regards,

on behalf of

Dr. Chenfeng Xiong

Academic Editor

PLOS ONE